# Utility of IL-6 in the Diagnosis, Treatment and Prognosis of COVID-19 Patients: A Longitudinal Study

**DOI:** 10.3390/vaccines10111786

**Published:** 2022-10-24

**Authors:** Vikram Jain, Pratap Kumar, Prasan Kumar Panda, Mohan Suresh, Karanvir Kaushal, Anissa A. Mirza, Rohit Raina, Sarama Saha, Balram J. Omar, Vivekanandhan Subbiah

**Affiliations:** 1Department of Internal Medicine (ID Division), AIIMS Rishikesh, Rishikesh 249203, India; 2Department of Biochemistry, AIIMS Rishikesh, Rishikesh 249203, India; 3Department of Microbiology, AIIMS Rishikesh, Rishikesh 249203, India

**Keywords:** cytokine storm, ferritin, LDH, D-dimer, SARS-CoV-2, coronavirus

## Abstract

COVID-19 has caused devastating effects worldwide ever since its origin in December 2019. IL-6 is one of the chief markers used in the management of COVID-19. We conducted a longitudinal study to investigate the role of IL-6 in diagnosis, treatment, and prognosis of COVID-19-related cytokine storm. Patients with COVID-19 who were admitted at AIIMS Rishikesh from March to December 2020 were included in the study. Patients with no baseline IL-6 value at admission and for whom clinical data were not available were excluded. Clinical and laboratory data of these patients were collected from the e-hospital portal and entered in an excel sheet. Correlation was seen with other inflammatory markers and outcomes were assessed using MS Excel 2010 and SPSS software. A total of 131 patients were included in the study. Of these, 74.8% were males, with mean age 55.03 ± 13.57 years, and mean duration from symptom onset being 6.69 ± 6.3 days. A total of 82.4% had WHO severe category COVID-19, with 46.56% having severe hypoxia at presentation and 61.8% of them having some comorbidity. Spearman rank correlation coefficient of IL-6 with D-dimer was 0.203, with LDH was −0.005, with ferritin was 0.3, and with uric acid was 0.123. A total of 11 patients received Tocilizumab at a mean duration from symptom onset of 18.09 days, and 100% mortality was observed. Deaths were reported more in the group with IL-6 ≥ 40 pg/mL (57.1% vs. 40.2%, *p* = 0.06). ICU admissions and ventilator requirement were higher in the IL-6 ≥ 40 pg/mL group (95.9% vs. 91.4%, *p* = 0.32 and 55.1% vs. 37.8%, *p* = 0.05). The study showed that IL-6 can be used as a possible “thrombotic cytokine marker”. Higher values of IL-6 (≥40 pg/mL) are associated with more deaths, ICU admissions, and ventilator requirement.

## 1. Introduction

COVID-19 is a viral illness caused by SARS-CoV-2 infection in humans. It is a respiratory pathogen, transmitted by close contact and droplets. The viral infection may be asymptomatic in up to 15.6% of individuals [1] and may cause symptoms in others. The main symptoms of the illness include fever, dry cough, and fatigue. Other symptoms include loss of taste, nasal congestion, sore throat, conjunctivitis, headache, myalgia, nausea, vomiting, diarrhoea, etc. [2] The disease has drawn worldwide attention ever since its first detection in December 2019 and, despite several efforts globally, it has spread across the world and has caused a health disaster. According to WHO, as of 8 July 2022, 5:33 pm, 551,226,298 confirmed cases and 6,345,595 deaths have been reported worldwide due to COVID-19 [3]. In India, the toll of confirmed COVID-19 cases has reached 43,585,554, with 525,343 deaths related to COVID-19 [3]. The disease has caused long-lasting effects in many patients—dyspnoea and psychiatric issues (anxiety, panic disorder, etc.).

Certain viruses, such as highly pathogenic SARS-CoV-2, influenza-associated viruses, and Ebola viruses, induce an excessive and prolonged cytokine/chemokine response known as “cytokine storms”, which results in high morbidity and mortality due to immunopathology [4]. Inflammatory markers, especially IL-6, CRP, Procalcitonin, and ESR, have been positively correlated with the severity of COVID-19 [5]. As of now, due to lack of definitive antiviral therapy against COVID-19, targeted immunotherapy has become treatment of choice in this massacre [6]. Out of all inflammatory markers, therapy against IL-6, Tocilizumab, reduces 28–30 day all-cause mortality, ICU admission, super infections, mechanical ventilation, and the combined endpoint of death or mechanical ventilation [7].

Due to lack of studies showing integration of all the major roles of IL-6 in COVID-19 illness, here, we performed a longitudinal study showing the role of IL-6 in diagnosis, treatment, and prognosis of COVID-19-related disease. This study will further strengthen our knowledge regarding IL-6 and COVID-19 disease.

## 2. Materials and Methods

The approval for this study was obtained from institute ethics committee of AIIMS Rishikesh with approval no. CTRI/2020/08/027169.

The study population is comprised of patients of age 18 years and above with diagnosis of COVID-19 (clinicoradiologically or positive COVID-19 RT-PCR testing of oropharyngeal and nasopharyngeal swabs) who were admitted at AIIMS Rishikesh during the 1st COVID-19 wave (from March 2020 to December 2020). The patients who did not have baseline IL-6 at admission and incomplete data in online e-hospital portal entry were excluded from the study.

All the demographic details, which included age, gender, and clinical details, such as chief complaints, comorbidities, various investigations, routine blood investigations, inflammatory markers, radiological investigations, treatment given, outcome (death or discharge), ICU stay, and mechanical ventilation, were obtained from the e-hospital portal of the institute and entered into an excel sheet at the timeline of 0, 2, 7, 14, 21, and 28 days from the duration of symptom onset. It was mandatory for all the patients to have a baseline IL-6 at the time of admission. The patients who had raised IL-6 received Tocilizumab based upon the clinician’s decision and availability.

The primary outcome was to assess the correlation of IL-6 with other inflammatory markers, to see the proportion of patients with raised IL-6 requiring ICU admission and mechanical ventilation, to look for death in the patients with raised IL-6, and to see the effect of IL-6 antagonist, Tocilizumab, on the outcome.

The data were entered in Microsoft excel 2010 and analyzed using excel and SPSS software. For outcome analysis, patients were divided into two groups—high IL-6 (IL-6 ≥ 40 pg/mL) and low IL-6 (IL-6 < 40 pg/mL). Spearman rank correlation coefficient was calculated to see how IL-6 values correlated with the other inflammatory markers. Chi-square test was used to see the statistical significance of outcome prediction with IL-6.

## 3. Results

We identified 213 patients, out of which 73 were excluded as baseline IL-6 data werenot available and, later on, 9 were excluded because of incomplete data in e-hospital portal. The remaining 131 patients were included in the study (Figure 1).

Baseline characteristics of all the patients are shown in Table 1. Mean duration from symptom onset at presentation was 6.69 + 6.3 days and, out of 131 patients, 82.4% patients had severe COVID-19 according to WHO and 46.5% of patients had severe hypoxia.

Largely four inflammatory biomarkers, ferritin, D-dimer, LDH, and uric acid, were measured and correlation was shown with IL-6 in the form of Spearman rank correlation coefficient. The spearman rank correlation coefficient of IL-6 with D-dimer was 0.203 (*n* = 43), with ferritin was 0.3 (*n* = 15), with LDH was −0.005 (*n* = 20), and with uric acid was 0.123 (*n* = 77). Figure 2 shows the scatterplot of IL-6 with inflammatory markers.

In hospital, death was considered as an important outcome assessor and compared in both the groups (Figure 3A–C). Death was reported more in patients with raised IL-6 compared with lower IL-6 values (57.1% vs. 40.2%, *n* = 131, *p* = 0.06). The need for ICU admission and mechanical ventilation was also higherin patients with raised IL-6, 95.9% vs. 91.2% (*n* = 131, *p* = 0.32) and 55.1% vs. 37.8% (*n* = 131, *p* = 0.054), respectively. The duration of hospital and ICU stay was higher in the group of raised IL-6 (22.4 vs. 18.6 days = 0.12 and 17.58 vs. 14.44 days = 0.16). IL-6 antagonist, Tocilizumab, was administered in 11 patients with raised IL-6 value after a mean duration from symptom onset of 18.09 days and death was reported in 100% of patients.

## 4. Discussion

Despite introduction of vaccines, COVID-19-related breakthrough infections are still high in various parts of the world [8]. It has become essential to discover markers of severe disease and treat them on an early basis to stop the incoming of new COVID-19-related massacre as we have experienced early [9]. IL-6 is among the few potential biomarkers whose level can predict the development of severe COVID-19 pneumonia [10] and its antagonist could help us prevent this if used within a given period of time [11,12].

In our knowledge, our study is first among all to correlate IL-6 with other inflammatory markers in COVID-19 and show its role in treatment and prognosis of the disease. Increase in inflammatory markers in viral infection is part of cytokine storm, which is characterized by hyper production of proinflammatory cytokines causing initiation of different signaling pathways and resulting in complicated medical symptoms, including fever, capillary leak syndrome, acute respiratory distress syndrome, and multiorgan failure, ultimately leading to death in the most severe cases [13]. Rise in levels of D-dimer is due to activation of inflammatory pathways secondary to imbalance between coagulation and fibrinolysis primarily in alveolus [14,15]. Our study showed mild positive correlation of IL-6 with D-dimer (R = 0.203). Level of LDH rises nonspecifically in the pathological condition of lungs, such asinflammation and cell damage [16]. Serum LDH levels have been used as an independent marker of COVID-19 severity and predictor of mortality in COVID-19 [17,18]. Our study showed a weak correlation of IL-6 with LDH (R = −0.005). Uric acid acts as an antioxidant scavenging oxygen free radicals and protecting cells from oxidative damages [19,20,21]. Levels of uric acid can be used as a prognostic marker in severe COVID-19 infection [22]. Our study showed weak positive correlation of IL-6 with serum uric acid levels (R = 0.123). Serum ferritin being an acute-phase reactant mirrors the degree of acute inflammation inside the body. A higher ferritin level indicates activation of monocyte–macrophage system in the body and it is responsive to alteration incytokines levels in blood [23]. Higher ferritin levels are associated with more severe disease and negative or poor outcome in COVID-19 disease [24]. Our study has shown weak correlation of IL-6 with ferritin (R = 0.3).

A raised level of serum IL-6 (cut-off—35 pg/mL) is associated with increased mortality and can be a useful prognostic marker in determining severity of COVID-19 [25]. IL-6 value of >37.5 pg/mL is associated with higher in-hospital mortality (sensitivity = 91.7% and specificity = 95.7%) [26]. Our study used a cut-off of 40 pg/mL for raised serum IL-6 levels and showed higher mortality in the raised IL-6 group (57.2% vs. 40.2%). Higher levels of IL-6 also correlate withmore requirement formechanical ventilation and ICU admission [27,28]. In our study, compared to the group of low IL-6, higher ICU admission (95.9% vs. 91.2%) and higher incidence of mechanical ventilation (55.1% vs. 37.8), along with prolonged ICU and hospital stay (17.58 vs. 14.44 days and 22.4 vs. 18.6 days, respectively) was found in patients with raised IL-6.

Addition of IL-6 antagonist is associated with absolute reduction in mortality of 4% [29]. When given within 24 h of ICU admission, treatment with IL-6 antagonists had resulted in higher incidence of hospital survival with both Tocilizumab and Sarilumab [30]. In 11 patients with raised IL-6 in our study, Tocilizumab was given based upon clinician’s decision at a mean duration from symptom onset of 18.09 days, and 100% mortality was observed.

Our study, however, suffers from some definite limitations. A loweramount of data of inflammatory markers turns out to be a major limitation. In our study, the administration of IL-6 antagonist, Tocilizumab, was delayed because of unclassified reasons, which reflect its negative impact on outcome. The sample size of this study further affects the outcome of the study to a certain extent.

## 5. Conclusions

IL-6 is an important inflammatory marker predicting severity of COVID-19. The finding of our study strengthens the positive correlation of IL-6 with D-dimer. Hence, IL-6 would be considered as a “Thrombotic cytokine storm marker”. Negligible correlation with LDH and uric acid strengthens its importance in COVID-19 illness. Findings of our study reinforce the role of IL-6 as an important prognostic marker in severe COVID-19. Moreover, negative results after giving Tocilizumab later during the course of illness may encourage its use during initial stages of disease. Our study provides evidence that a raised value of baseline IL-6 correlates with mortality.

## Figures and Tables

**Figure 1 vaccines-10-01786-f001:**
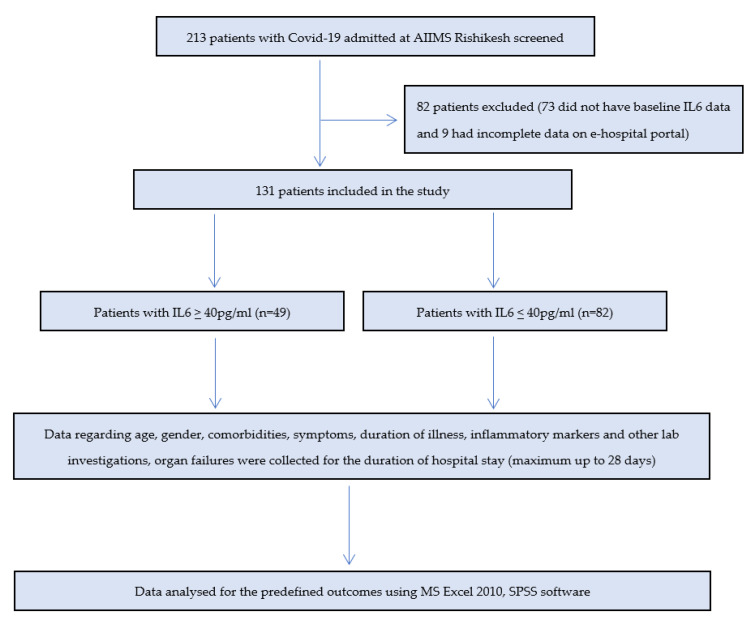
Flow of study.

**Figure 2 vaccines-10-01786-f002:**
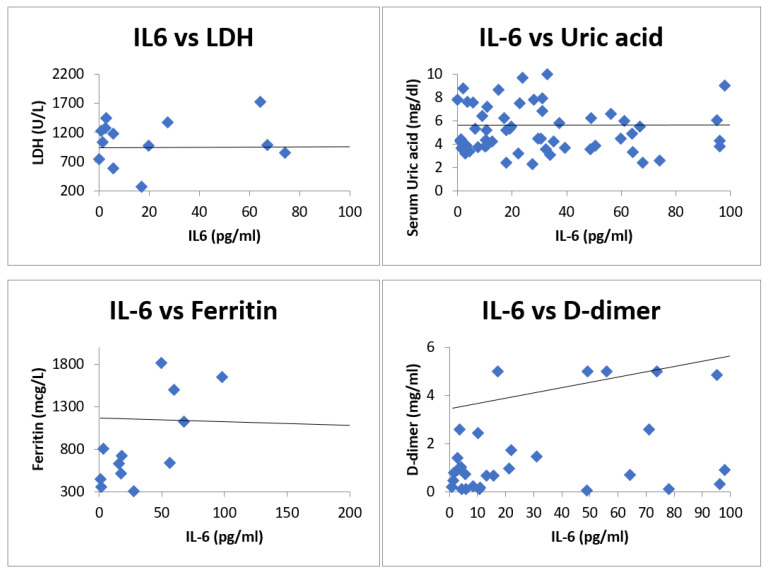
Scatterplots showing correlation of IL-6 with various inflammatory markers.

**Figure 3 vaccines-10-01786-f003:**
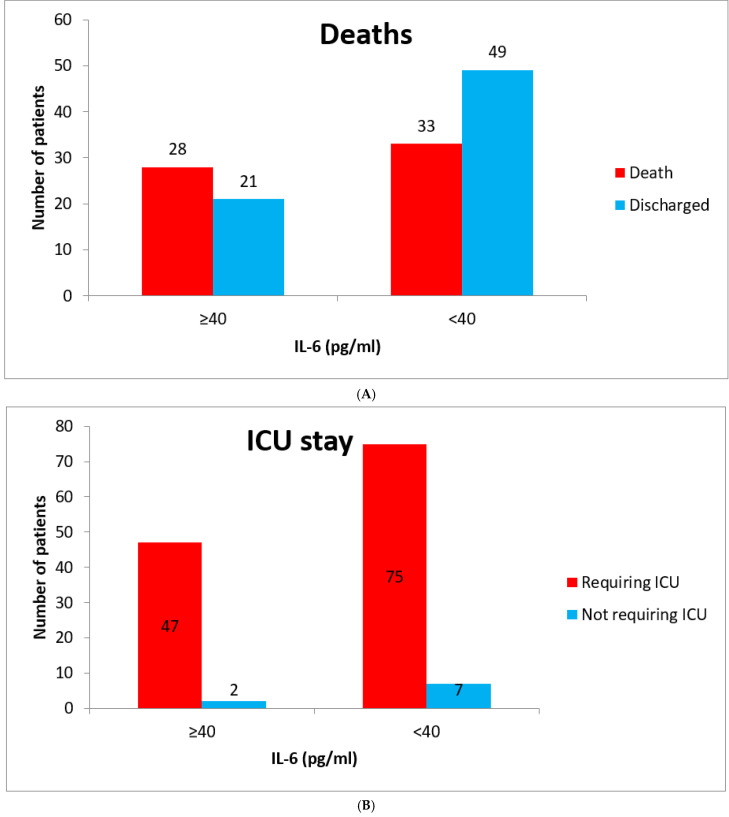
(**A**) Comparison of deaths between the groups. (**B**) Comparison of need for ICU stay between the groups. (**C**) Comparison of invasive mechanical ventilation (IMV) requirement between the groups.

**Table 1 vaccines-10-01786-t001:** General characteristics of the patients at admission (qCSI—quick COVID-19 severity score, P/F—PaO2/FiO2 ratio).

Total Subjects	131
Gender	
Male	98
Female	33
Mean age	55.03 ± 13.57 years
Mean duration from symptom onset at presentation	6.69 ± 6.3 days
Comorbidities	
None	50
Diabetes mellitus	56
Hypertension	46
Coronary artery disease	11
Chronic obstructive pulmonary disease	12
Cerebrovascular accident	2
Chronic kidney disease	3
Chronic liver disease	2
Autoimmune disease	5
Malignancy	3
Mean Charlson comorbidity index	2
Severity of COVID-19 at presentation (WHO)	
Mild	2
Moderate	21
Severe	108
Severity of COVID-19 at presentation (qCSI score)	
Low risk (0–3)	32
Low intermediate risk (4–6)	34
High intermediate risk (7–9)	1
High risk (10–12)	0
Respiratory failure at presentation	
No hypoxia	11
Mild hypoxia (P/F: 201–300)	13
Moderate hypoxia (P/F = 101–200)	41
Severe hypoxia (P/F = <100)	61

## Data Availability

These will be made available to others as required upon requesting from the corresponding author.

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
