# Peer review of "Utility of IL-6 in the Diagnosis, Treatment and Prognosis of COVID-19 Patients: A Longitudinal Study"

_vaccines, 2022, doi:10.3390/vaccines10111786_

Round 1

Reviewer 1 Report

This study shows that blood levels of IL-6 >40 pg/ml in COVID-19 infections might be a good 'thrombotic cytokine marker' for increased mortality, and inpatient ICU and ventilator use. Until now, many reports have been reported on the role of IL-6 in the pathogenesis of COVID-19 infection, and this study demonstrates its clinical importance. This is an important finding for the control of this viral infection, which is currently prevalent all over the world. However, it seems that it is necessary to change the format etc. for publication of this paper in the Journal.

[Major points]

1. Fig 1: Legend is necessary for reader comprehension.

2. Table 1: Could it be made a little more compact?

3. Fig 2: Legend is necessary for reader comprehension. Also, the D-dimer figure is out of alignment, so it needs to be corrected.

4. Fig 3: I think it would be better to show each panel A, B and C. And it would be good if Fig 3 Legend is necessary.

[Minor points]

1. Unify the concentration notation of IL-6. Depending on the location, it is different, such as ">40pg/ml" or ">40pg/ml".

2. Reference #37 is missing in Reference part. Also, check reference list one more time!

Author Response

REVIEWER’S COMMENT

RESPONSE

1.

Fig1: Legend is necessary for reader’s comprehension

Fig1 is a flow chart of the study. A higher resolution version of it has been inserted in the revised document and a figure description has been provided at the bottom for reader’s comprehension.

2.

Table1: Could it be made a little more compact

Changes made in the table

3.

Fig 2: Legend is necessary for reader comprehension. Also, the D-dimer figure is out of alignment, so it needs to be corrected.

The alignment of the D-dimer changed. The chart title changed to “IL-6 vs D-dimer”. The figure description has sufficient details for reader comprehension. Each points in scatterplot represent the value of the y-axis variable for the corresponding value of x-axis variable. Hence legend not mentioned.

4.

Fig 3: I think it would be better to show each panel A, B and C. And it would be good if Fig 3 Legend is necessary.

Legends present in the respective bar graphs. Chart title added in each graph for better comprehension and the type of graph changed to double bar graph.

5.

Unify the concentration notation of IL-6. Depending on the location, it is different, such as ">40pg/ml" or ">40pg/ml".

Changes made as per suggestion:

1.       In simple summary line number 8 and 9, “IL-6 more than 40pg/ml” changed to “IL-6 ≥40pg/ml”.

2.       In simple summary line number 11, “>40pg/ml” changed to “≥40pg/ml”.

3.       In Abstract line number 16 and 17, “IL-6 more than 40pg/ml” changed to “IL-6≥40pg/ml”.

6.

Reference #37 is missing in Reference part. Also, check reference list one more time!

Reference list checked and un-necessary items removed. [37] removed from line 45 of Discussion

Reviewer 2 Report

The inflammatory storm is an important transitional point from mild to severe and critical COVID-19 illness. Interleukin-6 (IL-6) is an important cytokine to induce inflammatory storm. In 2020, Tao Liu et al. have report that the dynamic change in IL-6 can be used as a marker for disease monitoring in patients with severe COVID-19 (EMBO Mol Med. 2020 Jul 7;12(7):e12421. doi: 10.15252/emmm.202012421.). In the same year, Jing Zhang et al. have also report that serum IL-6 is an indicator for severity in 901 patients with SARS-CoV-2 infection, while an IL-6 concentration higher than 37.65 pg/ml was predictive of in-hospital death (AUC 0.97 [95% CI 0.95–0.99], P < 0.001) with a sensitivity of 91.7% and a specificity of 95.7% (J Transl Med. 2020 Oct 29;18(1):406. doi: 10.1186/s12967-020-02571-x.). In contrast with these previous studies, the current study didn't provide more new ideas or knowledges.

Author Response

Respected XYZ,

Thanks for guidance.

The following changes have been made in the document as per the reviewers’ suggestion:

  1. In simple summary line number 8 and 9, “IL-6 more than 40pg/ml” changed to “IL-6 ≥40pg/ml”.
  2. In simple summary line number 11, “>40pg/ml” changed to “≥40pg/ml”.
  3. In Abstract line number 16 and 17, “IL-6 more than 40pg/ml” changed to “IL-6≥40pg/ml”.
  4. In introduction line number 14, “IAV” changed to “Influenza associated viruses”.
  5. In introduction line number 18, “PCT” changed to “Procalcitonin”.
  6. In material and methods line number 4, “RTPCR” changed to “RT-PCR”.
  7. Fig1 replaced with one with higher resolution.
  8. In Table 1; font reduced in size, T2DM changed to Diabetes mellitus, HTN changed to Hypertension, CAD changed to coronary artery disease, COPD changed to chronic obstructive pulmonary disease, CVA changed to cerebrovascular accident, CLD changed to chronic liver disease, CKD changed to chronic kidney disease. Also, “(qCSI- quick COVID severity score, P/F- PaO2/FiO2 ratio)” added in the table description.
  9. In Fig2; D-dimer alignment adjusted, chart title of “D-dimer” changed to “IL-6 vs D-dimer”. Also, the description of the figure changed to “Fig2: Scatterplots showing correlation of IL-6 with various inflammatory markers”.
  10. In Fig3A, chart title added “Deaths” and type of graph changed.
  11. In Fig3B, chart title added “ICU stay” and type of graph changed.
  12. In Fig3C, chart title added “IMV requirement”, type of graph changed and “invasive mechanical ventilation” added in the description.
  13. In line 45 of Discussion, “[37]” removed.
  14. In references, reference number 4,5,6,7,8,9 removed and renumbered accordingly.

Fig1 is the study flow diagram and can be understood by all as it shows the various steps in the study conducted.

Fig2 contains scatterplots showing the correlation of IL-6 with the various inflammatory markers. Each scatterplot has been labelled for readers understanding.

Fig3 has legends in the right side of the graph. Further chart title has been provided for readers understanding.

Thanks again,

Dr PK Panda (Corresponding author)

Reviewer 3 Report

The manuscript reported a longitudinal study analyzing the potential role of IL-6 in patients with COVID-19.

Although the study has several limitations, as the authors declared in the discussion, in my opinion any contribution to understand the mechanisms of this disease is very important.

I would like to ask authors to specify any acronym the first time it is introduced in the manuscript. This improves the whole readability, even to inexperienced readers. 

line 77 please write correctly RT-PCR.

Please replace Fig1 with one in higher resolution.

Round 2

Reviewer 2 Report

No further comments.